# Empowering Healthy Adolescents: A Dietary Adherence Tool Incorporating Environmental Factors Based on Korean Guidelines

**DOI:** 10.3390/nu17071102

**Published:** 2025-03-21

**Authors:** Jimin Lim, Soobin Lee, Ji-Yun Hwang, Jieun Oh

**Affiliations:** 1Department of Nutritional Science and Food Management, Ewha Womans University, Seoul 03760, Republic of Korea; jiminlim@ewha.ac.kr; 2Department of Clinical Nutrition, Ewha Womans University, Seoul 03760, Republic of Korea; 202cnc10@ewhain.net; 3Major of Foodservice Management and Nutrition, Sangmyung University, Seoul 03016, Republic of Korea; jiyunhk@smu.ac.kr; 4College of Science and Industry Convergence, Ewha Womans University, Seoul 03760, Republic of Korea

**Keywords:** adolescent, dietary guidelines for Koreans, adherence tool, social support, sustainable eating

## Abstract

**Background:** Adolescence is a critical period for adopting lifestyle behaviors that influence long-term health. While dietary habits are well-documented, the broader socio-cultural and environmental factors impacting these behaviors are underexplored. This study aimed to develop a dietary adherence tool for adolescents that aligns with the Dietary Guidelines for Koreans, incorporating individual and environmental factors for a comprehensive understanding of dietary behaviors. **Methods:** A nationwide survey was conducted with 1010 adolescents in Korea to develop and validate a dietary adherence tool based on the Dietary Guidelines for Koreans. Factor analyses and structural equation modeling confirmed the construct validity of the tool, and a grading system was established to evaluate adherence based on survey responses. **Results:** The survey included participants from 17 regions across South Korea. The original 22 candidate items were revised through factor analysis, resulting in the deletion of 4 items and the addition of 6 new items, leading to a final 24-item tool encompassing three domains: food intake, dietary and physical activity behaviors, and dietary culture. The validity of the revised tool remained intact. The mean dietary guideline adherence score of the participants was 54.5 (SD = 12.1), with domain scores of 39.1 (SD = 14.4) for food intake, 51.6 (SD = 16.6) for dietary and physical activity behaviors, and 66.8 (SD = 15.4) for dietary culture. **Conclusions:** The dietary adherence tool offers a comprehensive framework for assessing adolescent dietary behaviors by integrating food intake, dietary and physical activity behaviors, and environmental factors. By considering sustainability and family support, it promotes healthier and more sustainable eating patterns among adolescents.

## 1. Introduction

Adolescence is a period of active physical, internal, and emotional development that requires adequate and appropriate nutrition [1]. Unmet nutritional needs or imbalances during this stage can affect growth and increase the likelihood of developing chronic diseases in adulthood [2]. Nonetheless, globally, adolescent dietary patterns are characterized by high intakes of high-calorie ultra-processed foods, sodium, refined sugar, and saturated and trans fats; the frequent skipping of breakfast; a heavy reliance on fast food; and insufficient consumption of fruits and vegetables. These dietary patterns exert a significant influence on the escalating prevalence of adolescent overweight and obesity [3], a globally recognized public health crisis with profound implications for long-term health outcomes and socioeconomic burdens [4,5]. Furthermore, dietary behaviors established during adolescence tend to persist into adulthood, making early intervention crucial.

Environmental factors, including social supports from schools and families, play a key role in shaping healthy eating habits during adolescence [6]. Such support provides both practical and emotional reinforcement for making healthier dietary choices [7,8,9]. Interaction within the household environment shapes children’s eating behaviors [10,11], especially considering that parents, as primary caregivers, act as gatekeepers to their children’s dietary environment, influencing “what”, “when”, and “how” they eat [12]. The physical availability of food within the home is directly linked to dietary patterns; for example, greater availability of fruits and vegetables has been associated with increased consumption of healthy foods and reduced intake of energy-dense, nutrient-poor foods [13,14].

Dietary guidelines are recommendations for healthy eating that are designed to be easily understood and practiced in daily life by the general public [15]. Many countries have developed dietary guidelines tailored to their populations’ nutritional needs, cultural contexts, and dietary habits [16]. To assess adherence to these guidelines on country-specific contexts, various dietary indices have been developed, particularly for adolescents, whose dietary behaviors are crucial for long-term health [4,5]. Examples include the Dietary Guidelines Index [17], an assessment tool based on the 2013 Australian Dietary Guidelines; Norwegian Dietary Guideline Index [18], based on the Norwegian food-based dietary guidelines (FBDGs); Youth Healthy Eating Index [19], a tool that evaluates the Dietary Guidelines for Americans; and Dietary Quality Index for Adolescents [6], which is based on the FBDGs. However, these tools primarily focus on assessing food intake, often requiring detailed tracking of consumption quantity and frequency, which can be burdensome for individuals to record accurately and consistently [6,17,18,19]. Thus, there is a need for practical and scientifically validated tools that enable individuals to assess their dietary adherence independently and with minimal effort in everyday settings [20].

In Korea, the most recent revision of the dietary guidelines, the Dietary Guidelines for Koreans (2021), was developed based on scientific evidence, including national nutrition surveys, public health research, and an analysis of Korean dietary habits [15]. These guidelines are structured into nine sections categorized into three domains: food and nutritional intake, dietary and physical activity behaviors, and dietary culture [21]. The food and nutritional intake guidelines focus on chronic disease prevention, promoting the consumption of a balanced diet; a diet low in sodium, sugar, and saturated fatty acids; and sufficient water. The dietary and physical activity behaviors guidelines emphasize increased physical activity, breakfast consumption, and the avoidance of overeating and alcohol consumption to prevent obesity. The dietary culture guidelines advocate hygienic eating practices, encouraging the use of local foods and promoting environmental sustainability [15].

While these guidelines offer comprehensive recommendations, most previous research and dietary assessment tools developed for Koreans have primarily focused on food intake and nutritional adequacy or have been limited to individual-level factors, failing to align with the broader scope of the dietary guidelines [22,23,24,25,26]. Research based on the Dietary Guidelines for Koreans that considers both individual and environmental factors in shaping adolescent dietary behaviors remains limited.

To address these gaps, the present study seeks to develop an adolescent-friendly adherence tool designed to evaluate dietary behaviors in accordance with the Dietary Guidelines for Koreans. In pursuit of this objective, a survey was conducted among adolescents residing in Korea, with a focus on item selection and validation. Through this endeavor, we aim to contribute to the creation of a comprehensive assessment tool that integrates both individual and environmental factors, extending beyond the evaluation of food intake alone. Furthermore, the tool will examine the interaction between these factors in shaping healthy eating behaviors among adolescents.

## 2. Materials and Methods

### 2.1. Settings and Participants

A nationwide survey of adolescents was conducted between May and August 2023. Considering the population ratio between urban and rural areas, 25% of the respondents were recruited from “counties and below”, and the margin of error was set to <0.04 at the 95% confidence interval to ensure sampling at the national statistical quality level [27]. A total of 1010 adolescents residing in 17 provinces from five regions (Seoul, Gyeonggi, Chungcheong, Yeongnam, and Honam) who had consented to the purpose and procedures of the study were included. The study was approved by the Institutional Review Board (IRB) of Sangmyung University (IRB-SMU-C-2023-1-0015).

### 2.2. Survey Instrument

The online self-reported questionnaire used in this study was developed in accordance with the Dietary Guidelines for Koreans (DGK, 2021), as outlined in references [25,26,28,29,30,31,32,33]. The questionnaire comprises demographic data (age, sex, place of residence, weight, height, economic status), and body mass index (BMI) was calculated by dividing an individual’s weight in kilograms by the square of their height in meters (kg/m^2^), with height and weight derived from self-reported data. For each participant, BMI percentiles were standardized according to age- and sex-specific criteria, referencing the Korean National Growth Charts for children and adolescents, which were developed by the Korea Disease Control and Prevention Agency (KDCA) based on the WHO standards and adapted to reflect the growth patterns of the Korean population [34]. The questionnaire items categorized into three domains:Adolescent Food Intake. “How many servings of vegetables (e.g., greens, kimchi, mushroom, seaweed, laver, etc.) do you eat in a day?”, “How often do you eat yellow vegetables (e.g., pumpkin, carrots, paprika, etc.)?”, “How often do you eat a variety of food groups (e.g., vegetables, fruits, grains, meat, fish, eggs, legumes, and milk)?”, “How often do you eat fresh, whole fruits?”, “How often do you eat meat (e.g., beef, pork, chicken, etc.) or eggs (i.e., eggs, quail eggs, etc.) in total?”, “How often do you eat beans (e.g., tofu, soy milk, etc.) or nuts (e.g., peanuts, almonds, walnuts, etc.) in total?”, “How often do you eat dairy products such as milk (e.g., chocolate milk, strawberry milk, etc.), yogurt, and/or cheese in total?”, “How often do you drink sweetened beverages (e.g., soda, smoothies, bubble tea, fruit juice, etc.)?”, “How often do you eat processed meats derived from animal sources (e.g., ham, sausage, bacon, etc.)?”, and “How often do you eat deep-fried foods?”.Adolescent Dietary and Physical Activity Behaviors. “How often do you get at least 30 min of moderately vigorous exercise (e.g., brisk walking, cycling, badminton practice, school physical education classes, etc.)?”, “Do you control your weight to avoid being overweight or underweight?”, “How many meals do you eat per day?”, and “Do you eat regular meals at the same time every day?”.Adolescent Dietary Culture (in the context of environmental factors, including household and sustainability aspects). Household aspects: “How often were fresh fruits and vegetables available in your home (past 3 months)?”, “How often were fresh milk or dairy products available in your home (past 3 months)?”, “How often did your parents (caregivers) prepare meals for you (past 3 months)?”, “How often did your parents (caregivers) encourage you to eat healthy snacks (e.g., fruits, vegetables, milk, yogurt, etc.)?”. Sustainability aspects: “Do you check the expiration date (quality) of food?”, “Do you use your own dishes to serve the food kept in the refrigerator (leftovers)?”, “How often do you use locally produced agricultural products (local food)?”, and “How often do you use environmentally friendly products?”.

### 2.3. Questionnaire Validity and Reliability Analysis and Model Confirmation

Exploratory and confirmatory factor analyses were conducted to confirm the construct validity of the survey questionnaire [35,36,37]. First, factors and measurement variables (items) that fit the concepts and attributes of each domain were extracted via exploratory factor analysis, whereas non-fitting items were deleted. Moreover, structural equation modeling using AMOS 29.0 was employed to conduct confirmatory factor analysis to verify the fit of the model. Afterward, the structural equation models of the three domains were integrated, and further confirmatory factor analysis was performed to confirm the fit of the overall model. Weights were derived from the path coefficients of each domain’s scales in the structural equation model and used to calculate the scores for dietary guideline adherence tool items. In addition, we standardized the scores using data collected from the nationwide survey to establish a rating system (high, medium, or low) for the dietary guideline adherence tool.

For exploratory factor analysis, principal component analysis and the factor varimax method were applied to each domain, factors with an Eigen value > 1.0 were extracted, a factor loading criterion of 0.40 was used to categorize the factors, and items not meeting this criterion were deleted [38]. The Kaiser–Meyer–Olkin (KMO, ≥0.6) test, Bartlett’s test of sphericity (*p* < 0.05), and “total variance explained” were used to assess sample adequacy [39]. Confirmatory factor analysis using the best-fit structural equation model was estimated via maximum likelihood estimation, and model fit was evaluated using the following criteria: the root mean square error of approximation (RMSEA, ≤0.08) [40], standardized root mean square residual (SRMR, ≤0.05) [41], goodness-of-fit index (GFI, ≥0.9 and adjusted GFI [AGFI], ≥0.9), and comparative fit index (CFI, ≥0.9) [42,43]. The internal consistency of each domain within the dietary guideline adherence tool was assessed using Cronbach’s alpha reliability test.

### 2.4. Development of a Grading Criterion for the Dietary Guideline Adherence Tool

The response scores from the nationwide survey were coded according to the number of responses: 1 point for one to 5 points for five. The score for each question was calculated as follows: “(response score per question − 1) × 25” and converted to 100 points out of 100 for each question, and the score for each domain was calculated by multiplying the score for each question by the weight of each question in the domain. The weight of each item was derived by calculating the item weight through regression analysis. The total dietary guideline adherence score was calculated by multiplying the score for each domain by the weight of the selected domain and subsequently summing the three domain scores to yield a total score of 100. To provide a grading scale for the dietary guideline adherence tool, the scores from the national survey were standardized into percentile scores and divided into quartiles, with the >25 percentile being the highest, the 25–74.9 percentile being the middle, and the ≤24.9 percentile being the lowest.

### 2.5. Data Analysis

#### 2.5.1. Finalized Items for Assessing Dietary Guideline Adherence

A panel of experts (n = 6) (Appendix A Table A1) with at least 10 years’ experience in scale development and nutrition and diet-related research were asked to provide their opinions regarding the items in the Dietary Guidelines for Koreans, the domains to which they belong, and their weights.

#### 2.5.2. Statistical Analysis

All statistical analyses were performed using SPSS Statistics software (version 29.0.2; IBM, Armonk, NY, USA), and *p* values < 0.05 were considered statistically significant. Through expert panel consultation, the final 24-item tool was validated by comparing it to the revised 18-item version using the Independent Samples *T*-Test and Cohen’s Kappa Coefficient [44].

## 3. Results

As outlined in the workflow presented in Figure 1, this study culminates in the development of dietary guideline adherence tools that align with the DGK and expert recommendations.

### 3.1. Participants’ Demographic Characteristics

In total, 1010 individuals from the nationwide survey agreed to participate and completed all the questionnaires. The characteristics of the population are presented in Table 1. The study included 472 boys (46.7%) and 538 girls (53.3%) as participants, with mean ages of 16.0 (SD = 1.7) and 16.1 (SD = 1.6) years and a mean body mass index (BMI) of 22.6 (SD = 4.6) and 20.9 (SD = 4.1) kg/m^2^, respectively. In terms of educational level, two grade ranges participated: middle school (12–14 years of age, 28.4%, n = 287) and high school (15–18 years of age, 71.6%, n = 723). Regarding place of residence, 74.3% and 25.7% of participants resided in urban and rural (classified as towns, townships, or villages) areas, respectively.

### 3.2. Selection of Items and Final Scales for Each Dietary Guideline Domain

Table 2 shows the exploratory and confirmatory factor analysis results for the items in each of the food intake, dietary behavior, and dietary culture domains.

Of the ten candidate items in the food intake domain, three items, namely, “Frequency of processed meat intake”, “Frequency of fried food intake”, and “Frequency of sweetened beverage intake”, were removed from the exploratory factor analysis because they were inconsistent with theoretical expectations, and the remaining seven items were categorized into two factors. Factor 1 of the food intake domain included the following items: “Frequency of total vegetable intake”, “Frequency of yellow vegetable intake”, and “Intake frequency of various food groups”. Factor 2 comprised the following items: “Frequency of fresh, whole fruit intake”, “Frequency of meat or egg intake”, “Frequency of bean or nut intake”, and “Frequency of dairy product intake”. The goodness-of-fit test of the structural equation model revealed that the GFI, AGFI, RMSEA, and SRMR values of the food intake domain were 0.983, 0.964, 0.061, and 0.0339, respectively, indicating a “good” model fit.

The four items examined in the dietary and physical activity behaviors domain were categorized into two factors. “Frequency of vigorous exercise” and “Effort to maintain a healthy weight”—factors related to maintaining a healthy weight and exercise—were categorized into Factor 1, while “Frequency of meals in a day” and “Regularity of meals”—factors related to eating habits—were categorized into Factor 2. In the structural equation model fit test, the GFI, AGFI, RMSEA, and SRMR values were 0.996, 0.956, 0.088, and 0.0234, respectively, for the dietary and physical activity behaviors domain, indicating that the model’s fit was “good”, except for the RMSEA value.

Of the eight candidate items in the dietary culture domain, one item, “Frequency of using environmentally friendly products”, was removed owing to a strong correlation among items related to locally produced products (local food), and the remaining seven items were categorized into two factors. “Accessibility of fresh fruits/vegetables at home”, “Accessibility of fresh dairy products at home”, “Frequency of parental meal preparation”, and “Parental encouragement toward healthy snacking” were clustered into Factor 1. “Checking of expiration dates on food packaging”, “Use of one’s own dishes to serve leftovers”, and “Use of local food” were clustered into Factor 2. In the structural equation model fit test, the dietary culture domain yielded GFI, AGFI, RMSEA, and SRMR values of 0.982, 0.961, 0.064, and 0.0332, respectively, indicating that the model fit was favorable across all criteria. The CMIN/DF, GFI, AGFI, CFI, RMSEA, and SRMR values of the structural equation model for the three dietary guidelines combined were 3.79, 0.95, 0.93, 0.893, 0.053, and 0.048, respectively, signifying a “good” fit for all criteria.

Confirmatory factor analysis combining the models of the three dietary guideline domains revealed the following path coefficients (weights): food intake (0.35), dietary and physical activity behaviors (0.30), and dietary culture (0.35). The path coefficient value of each item in the integrated structural equation was used as the weight of the item upon calculating the dietary guideline adherence score. The Cronbach α values for the internal consistency of the items were 0.711, 0.473, and 0.713 for food intake, dietary and physical activity behaviors, and dietary culture, respectively, indicating that the reliability of food intake and dietary culture was adequate, while that of eating habits was marginally acceptable.

We included additional items based on the Dietary Guidelines for Koreans, as experts agreed that Guidelines 1 (“Eat a balanced diet comprising grains, meat, fish, eggs, legumes, and dairy products every day”), 2 (“Eat less salty, less sugary, and less fatty food”), and 3 (“Drink plenty of water”) were not sufficiently represented. The six additional items comprised “Frequency of grain intake”, “Frequency of fish or seafood intake”, “Frequency of spicy and salty soup intake”, “Frequency of sweetened beverage intake”, “Frequency of fried food intake”, and “Frequency of drinking water”, and the validity of the final scale was reviewed based on the analysis results before and after collecting expert opinions. The validity of the final 24-item tool, expanded to include 6 additional food intake domain items to evaluate dietary guideline adherence, was maintained (Table 3 and Table 4).

### 3.3. Dietary Guideline Adherence Scores and Grading Criteria for Adolescents

Table 5 and Appendix A Table A2 present the final dietary guideline adherence checklist items derived from the data of adolescents (n = 1010) who participated in the national survey.

The mean dietary guideline adherence score of the national survey participants was 54.5 (SD = 12.1). The food intake, dietary and physical activity behaviors, and dietary culture domains yielded scores of 39.1 (SD = 14.4), 51.6 (SD = 16.6), and 66.8 (SD = 15.4), respectively, with dietary culture exhibiting the highest mean score. The domain scores were standardized to percentiles, and the cut-off scores for the top 25th percentile (good dietary practice) were ≥62.6, ≥48.1, ≥63.0, and ≥78.0 for the total score, food intake domain, dietary and physical activity behaviors domain, and dietary culture domain, respectively. In contrast, the cut-off scores for the bottom 25th percentile (poor dietary adherence) were ≤45.8, ≤28.7, ≤40.4, and ≤56.4 for the total score, food intake domain, dietary and physical activity behaviors domain, and dietary culture domain, respectively, with the dietary culture domain yielding the highest cut-off score among the three dietary adherence tool domains (Table 6).

## 4. Discussion

The developed adherence tool represents a significant advancement in assessing adolescents’ adherence to the Dietary Guidelines for Koreans, particularly by integrating not only food intake but also cultural and environmental factors that shape healthy dietary behaviors. Existing dietary assessment tools, such as the food intake frequency survey [22], Korean Healthy Eating Index (KHEI) [45], and the Korean Dietary Quality Index (KDQI) [24], offer the advantage of evaluating diet quality through precise dietary data; however, they primarily concentrate on food and nutrient intake and lack consideration of environmental influences. While these tools are valuable for assessing nutrient intake, they do not comprehensively evaluate the broader context in which dietary choices are made. While the Youth Food Consumption Behavior Survey from the Korea Rural Economic Research Institute [46] and the Youth Health Behavior Survey from the Korean Health Promotion Agency [47] provide valuable insights into adolescent food preferences and health behaviors, they do not address cultural relevance or environmental sustainability. The youth nutrition index [23,25,26], developed by the Ministry of Food and Drug Safety, evaluates the dietary quality and eating behaviors of Korean adolescents. It classifies participants into three groups—“risk”, “possible risk”, and “good”—based on quartile scores and is used to assess dietary behaviors easily. However, it does not consider water intake, environmental factors, or cultural aspects of eating, which are emphasized in the Korean Dietary Guidelines. While many dietary assessment tools focus on food and nutrient intake, they are neither fully consistent with dietary guidelines nor sufficiently thorough in capturing environmental influences.

### 4.1. Evaluation of Checklist Alignment with Korean Dietary Guidelines

In contrast, the tool developed in this study effectively addresses these gaps by aligning with the Dietary Guidelines for Koreans. When examining the final selected checklist for the food intake domain, the intake of specific food groups—including total vegetables, yellow vegetables, fresh and whole fruits, beans and nuts, fish and seafood, meat and eggs, and dairy products—aligned with **Guideline 1**, that is, “Eat a balanced diet comprising grains, meat, fish, eggs, legumes, and dairy products every day”, providing valuable insights into the intake of diverse food groups and their contribution to overall meal composition. **Guideline 2** included three questions on the consumption of spicy and salty soups, sweetened beverages, and fried foods, with the intent to “Eat less salty, less sugary, and less fatty food”, and one question was on drinking water under **Guideline 3**, “Drink plenty of water”. For the dietary and physical activity behaviors domain, **Guideline 4**, “Avoid binge eating or over-eating and increase physical activity to maintain a healthy weight”, included two questions regarding efforts to maintain a healthy weight and frequency of vigorous exercise. **Guideline 5**, “Do not skip breakfast”, included two questions concerning the number of meals consumed per day and meal regularity. For the dietary culture domain, **Guideline 6**, “Prepare food hygienically and only as much as required”, one question was selected regarding checking food expiration dates. Regarding **Guideline 7**, “Practice portion control when eating”, one question was selected concerning using one’s own plate to serve leftovers. For **Guideline 8**, “Do not drink alcohol”, related items were excluded from the assessment during the first phase of the survey and for underage respondents. **Guideline 9**, “Enjoy local foods and eat with respect for the environment”, consists of environmental aspects, comprising social support and sustainability, such as eco-friendliness. To evaluate these factors, four items were selected: whether the user had access to locally produced foods, the availability of fresh fruits, vegetables, and dairy products at home; whether their parents prepared their meals; and whether their parents encouraged them to eat healthy snacks. The concept of sustainable nutrition emphasizes a healthy diet that is culturally acceptable, easily accessible, and environmentally friendly [48]. This integration provides a more comprehensive assessment of dietary adherence, reflecting the socio-cultural and environmental contexts in which adolescents live.

### 4.2. Ensuring a Supportive Food Environment for Adolescents

Another key strength of the tool lies in its ability to measure not only individual food choices but also the environmental influences that significantly shape those choices. Among these influences, sustainability has gained increasing global recognition, with proactive movements in the food sector drawing significant attention. As the need for sustainable diets—those that are not only healthy but also environmentally responsible—continues to grow, integrating sustainability-related questions, such as eco-conscious behaviors and the consumption of locally sourced foods, enhances the comprehensiveness of dietary assessment tools.

The United Nations has highlighted the critical connection between food production, consumption, and access to healthy food under Goal 2 of the Sustainable Development Goals: “End hunger, achieve food security and improved nutrition, and promote sustainable agriculture” [49]. To further advance sustainable food systems, the World Health Organization has proposed policies and action plans, such as updating dietary guidelines to incorporate sustainability and encouraging diverse, healthy diets [50]. Aligned with these global initiatives, Korea’s Ministry of Agriculture, Food, and Rural Affairs has emphasized the importance of sustainable diets in the Third Basic Plan for Dietary Education (2020), underscoring their role in achieving multiple social values, including agricultural sustainability, environmental protection, and public health [51].

In addition to these broader sustainability efforts, household support is another critical factor in shaping dietary adherence. Family dynamics, particularly parental influence on meal preparation and the availability of healthy food have been shown to have a considerable impact on adolescents’ eating behaviors [52,53,54,55,56]. The inclusion of these factors provides valuable insights into how social and cultural contexts influence dietary adherence. Moreover, ensuring adherence to sustainable dietary practices requires active motivation within social and cultural contexts [57]. In particular, parental support plays a crucial role in shaping children’s values, influencing attitudes and behaviors, and fostering long-term health and well-being [58,59,60]. Internalized motivation, often reinforced through family influence, further strengthens adherence to sustainable dietary choices [61]. Additionally, the broader food system, as a form of social support, is essential for ensuring that adolescents have consistent access to sustainable food options [62]. Therefore, fostering a supportive environment—both within families and society—is vital for promoting sustainable dietary habits among youth.

### 4.3. Strengths and Limitations

The tool developed in this study is significant because it is the first to comprehensively assess the three domains of food intake, dietary behaviors, and dietary culture based on the Korean Dietary Guidelines, reflecting on adolescent food consumption characteristics and the influence of environmental factors such as social support and environmental sustainability on healthy eating. In addition, the validity and reliability of the assessment questions were improved through nationwide surveys and step-by-step opinions from experts, and an assessment rating system was established by weighting each domain and question. Finally, while most assessment tools for adolescents present difficulty when self-assessing their diet and interpreting the results without expert intervention, the tool developed in this study is a user-friendly scale that allows adolescents to easily assess their diet and verify their dietary status.

Despite its significance, a limitation of this study’s tool is that it was developed to assess adolescent dietary habits in Korea, and these habits vary significantly during the school year and vacation [63]; therefore, a more detailed scale that takes this into account is warranted. In addition, numerous adolescents acquire food information through online media, such as YouTube and social media [64,65], and some studies have raised concerns regarding the influence of such media on dietary behaviors [66]. Notably, prior research has even suggested that increased media consumption is associated with a higher intake of ultra-processed foods [67]. However, this study did not include specific items to assess the consumption of ultra-processed foods, which represents a limitation. Future research could enhance the comprehensiveness and precision of the tool by incorporating items related to ultra-processed food consumption and by further investigating the effects of media exposure, including the frequency and duration of use, content viewed, and self-regulation over media consumption, on adolescents’ eating habits.

A second limitation lies in the self-reported nature of the tool, which may introduce bias into the results [68]. While this was a deliberate design choice to make the tool more user-friendly and accessible, we recognize that it can limit accuracy, particularly in food intake and economic status, where adolescents subjectively estimate their standing. To address this, future studies will aim to establish standardized national criteria for evaluating economic status and provide more detailed guidance on serving sizes and other relevant factors to improve the accuracy of these assessments.

The third limitation is related to the frequency of responses in the questionnaire, which was standardized across most food items. As the response frequencies were adapted from an existing survey, this approach was chosen to reduce respondent burden and simplify the response process. However, it may not fully reflect the specific consumption patterns of different food groups. While this simplification aimed to ease participation, it may not adequately capture the variation in consumption frequency among different food groups. Future studies should explore the potential to refine this aspect of the tool to improve the accuracy and precision of dietary assessments.

Finally, while the tool establishes a baseline for dietary adherence, its long-term impact on dietary behavior has yet to be explored. Future research should investigate whether this tool can track and influence dietary changes over time, providing insights into its effectiveness in promoting sustained adherence to dietary guidelines and healthy eating habits. Moreover, future research could explore the development and evaluation of concrete intervention programs based on this tool to support adolescents in practicing sustainable eating habits. Such interventions might include eco-friendly food purchasing guides or local food consumption promotion campaigns, which would contribute not only to health but also to environmental sustainability.

## 5. Conclusions

The developed dietary adherence tool marks a significant step forward in evaluating adolescent dietary behaviors by integrating food intake, dietary and physical activity behaviors, and dietary culture in alignment with the Dietary Guidelines for Koreans. Unlike existing assessment methods, this tool expands beyond nutrient intake to capture the broader socio-cultural and environmental influences on dietary choices. By incorporating sustainability-related questions and considering family and social support systems, the tool provides a more comprehensive framework for assessing and promoting healthier, more sustainable eating patterns among adolescents.

As global attention toward sustainable food systems continues to grow, the relevance of tools that assess and encourage environmentally responsible dietary behaviors becomes increasingly vital. This study highlights the need for dietary assessment instruments that are not only scientifically rigorous but also practical and adaptable to the evolving dietary landscape. While this tool offers valuable insights, future refinements should consider seasonal variations in adolescent eating patterns and the influence of digital media on dietary choices, such as the impact of ultra-processed food consumption, and the potential bias introduced by self-reported data, particularly in food intake and economic status. Expanding its applicability across different contexts will further enhance its potential to support adolescents in making informed, health-conscious, and environmentally sustainable dietary decisions. Ultimately, this tool serves as a foundation for advancing adolescent nutrition research and fostering healthier food environments that align with both public health and sustainability goals.

## Figures and Tables

**Figure 1 nutrients-17-01102-f001:**
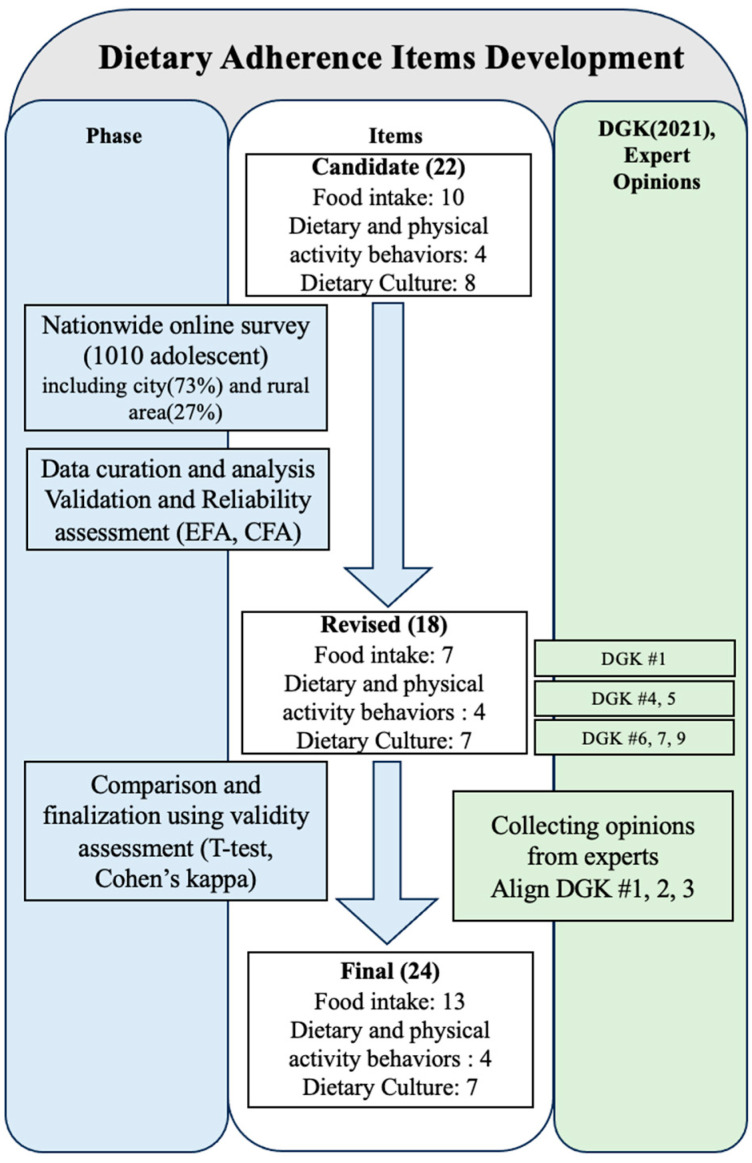
Workflow for final phase of dietary guideline adherence item development in line with DGK and expert recommendations.

**Table 1 nutrients-17-01102-t001:** Participants’ demographic characteristics, Korea, 2023 (n = 1010).

		12–14 yrs ^1^ n = 287 (28.4%)		15–18 yrs ^2^ n = 723 (71.6%)		Total n = 1010 (100%)	
		Boys n = 145 (50.5%)	Girls n = 142 (49.5%)	Boys n = 327 (45.2%)	Girls n = 396 (54.8%)	Boys n = 472 (46.7%)	Girls n = 538 (53.3%)
Age	Age (years)	13.8 ± 0.4	13.9 ± 0.5	16.9 ± 1.1	16.9 ± 1.1	16.0 ± 1.7	16.1 ± 1.6
Anthropometric data	Weight (kg) ^3^	66.3 ± 16.8	52.7 ± 12.5	68.5 ± 14.1	55.0 ± 11.3	67.8 ± 15.0	54.4 ± 11.7
	Height (cm)	172.0 ± 6.5	160.6 ± 4.9	173.6 ± 6.6	161.4 ± 5.2	172.2 ± 7.0	161.2 ± 5.2
	BMI (kg/m^2)^	22.3 ± 4.8	20.4 ± 4.6	22.7 ± 4.5	21.1 ± 3.9	22.6 ± 4.6	20.9 ± 4.1
		N (%)	N (%)	N (%)	N (%)	N (%)	
	Underweight ^4^	27 (18.6)	47 (33.1)	38 (11.6)	101 (25.6)	213 (21.1)	
	Normal	72 (49.7)	75 (52.8)	162 (49.6)	206 (51.8)	515 (51)	
	Overweight	15 (10.3)	12 (8.4)	50 (15.3)	42 (10.7)	119 (11.8)	
	Obesity	31 (21.4)	8 (5.7)	77 (23.5)	47 (11.9)	163 (16.1)	
		N	%	N	%	N (%)	
Place of residence	Seoul Capital City	29	10.1	84	11.6	113 (11.2)	
	Busan Metropolitan City	16	5.6	31	4.3	47 (4.7)	
	Daegu Metropolitan City	14	4.9	44	6.1	58 (5.7)	
	Incheon Metropolitan City	13	4.5	34	4.7	47 (4.7)	
	Gwangju Metropolitan City	7	2.4	21	2.9	28 (2.8)	
	Daejeon Metropolitan City	9	3.1	19	2.6	28 (2.8)	
	Ulsan Metropolitan City	9	3.1	25	3.5	34 (3.4)	
	Sejong Special Self-Governing City	7	2.4	6	0.8	13 (1.3)	
	Gyeonggi-do	83	28.9	185	25.6	268 (26.5)	
	Gangwon-do	8	2.8	19	2.6	27 (2.7)	
	Chungcheongbuk-do	11	3.8	27	3.7	38 (3.8)	
	Chungcheongnam-do	19	6.6	45	6.2	64 (6.3)	
	Jeollabuk-do	9	3.1	29	4	38 (3.8)	
	Jeollanam-do	18	6.3	35	4.8	53 (5.2)	
	Gyeongsangbuk-do	14	4.9	41	5.7	55 (5.4)	
	Gyeongsangnam-do	17	5.9	67	9.3	84 (8.3)	
	Jeju Special Self-Governing Province	4	1.4	11	1.5	15 (1.5)	
Research site	City area	209	72.8	541	74.8	750 (74.3)	
	Rural area ^5^	78	27.2	182	25.2	260 (25.7)	
Economic Status ^6^	High	32	11.1	45	6.2	77 (7.6)	
	Upper-middle	89	31	170	23.5	259 (25.6)	
	Middle	115	40.1	380	52.6	495 (49)	
	Lower-middle	45	15.7	106	14.7	151 (15)	
	Low	6	2.1	22	3	28 (2.8)	

^1^ Middle school participants in Korea; ^2^ High school participants in Korea; ^3^ Weight and height are self-reported; ^4^ Nutritional status is categorized according to BMI percentiles: a BMI under the 5th percentile indicates underweight, between the 5th and 85th percentiles indicates normal, the 85th to 95th percentiles indicates overweight, and at or above the 95th percentile indicates obesity; ^5^ The administrative divisions of the Republic of Korea include eup (towns), myeon (townships), and ri (villages) as rural units under provincial and city jurisdictions; ^6^ Economic status is self-reported.

**Table 2 nutrients-17-01102-t002:** Exploratory and confirmatory factor analyses of the nationwide survey questionnaire.

Exploratory Factor Analysis					Confirmatory Factor Analysis
Food Intake	Guideline ^(1)^	Item	Factor 1	Factor 2	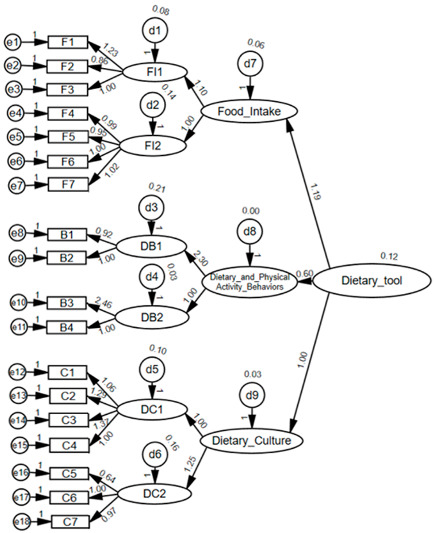 CMIN/DF = 3.79, GFI = 0.95, Adjusted GFI = 0.93, CFI = 0.893, RMSEA = 0.053, SRMR = 0.048
	1	Frequency of total vegetable intake (F1)	0.788	-	
	1	Frequency of yellow vegetable intake (F2)	0.781	-	
	1	Intake frequency of various food groups (F3)	0.651	-	
	1	Frequency of fresh, whole fruit intake (F4)	-	0.509	
	1	Frequency of meat or egg intake (F5)	-	0.706	
	1	Frequency of bean or nut intake (F6)	-	0.618	
	1	Frequency of dairy product intake (F7)	-	0.738	
		Eigen value Cumulative variance (%)	1.83 26.10	1.79 25.61	
	KMO = 0.781, Bartlett’s test of sphericity χ^2^ = 1043.15 (*p* < 0.001)				
Dietary and Physical Activity Behaviors	Guideline ^(1)^	Item	Factor 1	Factor 2	
	4	Frequency of vigorous exercise (B1)	0.742	-	
	4	Effort to maintain a healthy weight (B2)	0.826	-	
	5	Frequency of meals in a day (B3)	-	0.867	
	5	Regularity of meals (B4)	-	0.691	
		Eigen value Cumulative variance (%)	1.33 33.17	1.25 31.24	
	KMO = 0.570, Bartlett’s test of sphericity χ^2^ = 228.07 (*p* < 0.001)				
Dietary Culture	Guideline ^(1)^	Item	Factor 1	Factor 2	
	6	Accessibility of fresh fruits/vegetables at home (C1)	0.707	-	
	6	Accessibility of fresh dairy products at home (C2)	0.701	-	
	6	Frequency of parental meal preparation (C3)	0.701	-	
	6	Parental encouragement toward healthy snacking (C4)	0.766	-	
	6	Checking of expiration dates on food packaging (C5)	-	0.712	
	7	Use of one’s own dishes to serve leftovers (C6)	-	0.738	
	9	Use of local food (C7)	-	0.670	
		Eigen value Cumulative variance (%)	2.14 30.51	1.62 23.14	
	KMO = 0.767, Bartlett’s test of sphericity χ^2^ = 1177.04 (*p* < 0.001)				

^(1)^ DGK, Dietary Guidelines for Koreans: (1) Eat a balanced diet comprising grains, meat, fish, eggs, legumes, and dairy products every day; (2) Eat less salty, less sugary, and less fatty food; (3) Drink plenty of water; (4) Avoid binge eating or over-eating and increase physical activity to maintain a healthy weight; (5) Do not skip breakfast; (6) Prepare food hygienically and only as much as required; (7) Practice portion control when eating; (8) Do not drink alcohol; (9) Enjoy local foods and eat with respect for the environment; KMO, Kaiser–Meyer–Olkin; CMIN/DF, minimum discrepancy of confirmatory factor analysis/degrees of freedom; GFI, goodness-of-fit index; CFI, comparative fit index; RMSEA, root mean square error of approximation; SRMR, standardized root mean square residual.

**Table 3 nutrients-17-01102-t003:** Validation of final checklist items for the dietary guideline adherence tool.

Total (N = 1010)	18 Items ^1^	24 Items	t	*p*
Food intake (factor 1)	43.6 ± 20.7	43.6 ± 20.7	0	1
Food intake (factor 2)	43.1 ± 19.7	43.1 ± 19.7	0	1
Total Food intake score	43.3 ± 17.2	46.4 ± 17.3	4.06 ^3^	0.001
Dietary and physical activity behaviors (Factor 1)	45.7 ± 23.4	45.7 ± 23.4	0	1
Dietary and physical activity behaviors (Factor 2)	57.1 ± 19.1	57.1 ± 19.1	0	1
Total Dietary and physical activity behaviors score	51.7 ± 16.6	51.7 ± 16.6	0	1
Dietary culture (Factor 1)	63.7 ± 17.0	63.7 ± 17.0	0	1
Dietary culture (Factor 2)	70.6 ± 18.6	70.6 ± 18.6	0	1
Total Dietary culture score	67.0 ± 15.4	67.0 ± 15.4	0	1
DGA ^2^ score	54.1 ± 12.7	54.7 ± 12.1	0.95	0.344

^1^ Data are shown as the mean ± SD; ^2^ DGA, dietary guideline adherence; ^3^
*p* < 0.001, statistically significant.

**Table 4 nutrients-17-01102-t004:** Comparison of the reliability of dietary guideline adherence grades between the 18-item and 24-item scales.

		DGA ^1^ Grades (24-Item)				Total
			Excellent (≥75th)	Fair (25–74.9th)	Poor (<25th)	
DGA grades (18-item)	Excellent (≥75th)	N	252	4	0	256
		%	25.0	0.4	0	25.4
	Fair (25–74.9th)	N	6	476	1	483
		%	0.6	47.1	0.1	47.8
	Poor (<25th)	N	0	25	246	271
		%	0	2.4	24.4	26.8
Total		N	258	505	242	1010
		%	25.6	49.9	24.5	100
Symmetric measures			SE	T	*p*	
Kappa coefficient		0.943	0.009	41.71	<0.001	

^1^ DGA, dietary guideline adherence.

**Table 5 nutrients-17-01102-t005:** Final dietary guideline adherence tool checklist items for adolescents.

Food Intake (13) ^1^	Dietary and Physical Activity Behaviors (4)	Dietary Culture (7)
1. Frequency of total vegetable intake	1. Frequency of vigorous exercise	1. Accessibility of fresh fruits/vegetables at home
2. Frequency of yellow vegetable intake	2. Effort to maintain a healthy weight	2. Accessibility of fresh dairy products at home
3. Intake frequency of various food groups	3. Frequency of meals in a day	3. Frequency of parental meal preparation
4. Frequency of fresh, whole fruit intake	4. Regularity of meals	4. Parental encouragement toward healthy snacking
5. Frequency of meat or egg intake		5. Checking of expiration dates on food packaging
6. Frequency of bean or nut intake		6. Use of one’s own dishes to serve leftovers
7. Frequency of dairy product intake		7. Use of local food
8. Frequency of grain intake		
9. Frequency of fish or seafood intake		
10. Frequency of drinking water		
11. Frequency of sweetened beverage intake ^2^*		
12. Frequency of spicy and salty soup intake ^2^*		
13. Frequency of fried food intake ^2^*		

^1^ Participants were asked to rate their agreement with statements on a 5-point Likert scale. ^2^* Reverse-coded.

**Table 6 nutrients-17-01102-t006:** Evaluation criteria on dietary guideline adherence tool checklist items for adolescents.

Group	Score (N = 1010)	Evaluation Criteria		
		Excellent (75–100%)	Fair (25–74.9%)	Poor (0–24.9%)
DGA ^1^ score	54.5 ± 12.1 ^2^	62.6–100	45.9–62.5	0–45.8
Food intake	39.1 ± 14.4	48.1–100	28.8–48.0	0–28.7
Dietary and physical activity behaviors	51.6 ± 16.6	63.0–100	40.5–62.9	0–40.4
Dietary culture	66.8 ± 15.4	78.0–100	56.5–77.9	0–56.4

^1^ DGA, dietary guideline adherence; ^2^ Data are shown as the mean ± SD.

## Data Availability

Access to data can be requested from the corresponding author due to privacy reasons.

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
