# Peer review of "Empowering Healthy Adolescents: A Dietary Adherence Tool Incorporating Environmental Factors Based on Korean Guidelines"

_nutrients, 2025, doi:10.3390/nu17071102_

Round 1

Reviewer 1 Report

Comments and Suggestions for Authors

The manuscript by Lim et al. is based on a very important area, improving eating habits in adolescents with a view to promote life long healthy and balanced eating habits. Thus, the authors developed a dietary adherence tool reflecting and representing the advice provided in the Dietary Guidelines for Koreans, which includes both individual and environmental factors for a comprehensive understanding of dietary behaviours. However, before valid conclusions can be drawn, the methodological points below should be addressed by the authors: 

Methods

  • Since this is a self-report online questionnaire, how was height and weight and BMI calculated. Is there a system in Korea where this is measured by professionals and documented so the adolescents know their results, or is this based on subjective self- assessment?
  • Also, BMI categories for teens are based on sex-specific BMI-for-age percentiles (or BMI percentiles). Can the authors please address this as it appears only the BMI has been reported.
  • 2 Survey instrument: were any questions asked about processed foods in general (apart from the question about processed meats)? What about vegetarian/vegan products?
  • 2 Survey instrument: The heading “Adolescent eating habits” is followed by a question about physical activity. Then questions about weight stability and number of means a day etc are asked. Please use a different heading that captures the content of the questions asked.

Results

  • The frequency of consumption of the different food items should differ based on what food item is being asked about. The authors should provide more information (possibly in the methods section) on these scales of frequency of consumption as using one broad scale for all items can lead to inaccuracies in the dietary data collected.

Author Response

comments 1: The manuscript by Lim et al. is based on a very important area, improving eating habits in adolescents with a view to promote life long healthy and balanced eating habits. Thus, the authors developed a dietary adherence tool reflecting and representing the advice provided in the Dietary Guidelines for Koreans, which includes both individual and environmental factors for a comprehensive understanding of dietary behaviours. However, before valid conclusions can be drawn, the methodological points below should be addressed by the authors: 

Methods

Since this is a self-report online questionnaire, how was height and weight and BMI calculated. Is there a system in Korea where this is measured by professionals and documented so the adolescents know their results, or is this based on subjective self- assessment?

Also, BMI categories for teens are based on sex-specific BMI-for-age percentiles (or BMI percentiles). Can the authors please address this as it appears only the BMI has been reported.

2 Survey instrument: were any questions asked about processed foods in general (apart from the question about processed meats)? What about vegetarian/vegan products?

2 Survey instrument: The heading “Adolescent eating habits” is followed by a question about physical activity. Then questions about weight stability and number of means a day etc are asked. Please use a different heading that captures the content of the questions asked.

Results

The frequency of consumption of the different food items should differ based on what food item is being asked about. The authors should provide more information (possibly in the methods section) on these scales of frequency of consumption as using one broad scale for all items can lead to inaccuracies in the dietary data collected.

Response 1Thank you very much for your thoughtful and constructive feedback. I truly appreciate the time and effort you dedicated to reviewing the manuscript. Your comments have been invaluable in enhancing the quality of this work. Below are the changes made in response to your insightful feedback:

- First Comment: I sincerely appreciate your insightful feedback regarding the anthropometric data. In this study, height and weight were self-reported by participants through an online questionnaire, and BMI was calculated based on these self-reported values. To address this, I have provided additional details on BMI calculations and the criteria based on sex-specific BMI-for-age percentiles. BMI was calculated by dividing an individual’s weight in kilograms by the square of their height in meters (kg/m²). For each participant, BMI percentiles were standardized according to age- and sex-specific criteria, referencing the Korean National Growth Charts for children and adolescents, which were developed by the Korea Disease Control and Prevention Agency (KDCA) based on the WHO standards and adapted to reflect the growth patterns of the Korean population (P.3, L.112-118). Additionally, I have included the classification results for underweight, normal weight, overweight, and obesity in Table 1 (P.6, L.213). Furthermore, I have added a note under the table to clarify that "Height and weight were self-reported" and to specify the nutritional status classification: "A BMI between the 5th and 85th percentiles indicated normal weight, the 85th to 95th percentiles indicated overweight, and a BMI at or above the 95th percentile indicated obesity" (P.6, L.214-217). Thank you again for your thoughtful comment, which helped refine the clarity and accuracy of the anthropometric data section.

-Second Comment: I completely agree with your second point regarding the survey instruments. Based on the most frequently consumed foods in Korea, we specifically selected processed meat only, without including general processed foods or vegetarian/vegan products. Additionally, we provided more details by clarifying that it is "derived from animal sources" (P.3, L.130). Additionally, as you suggested, I have changed the heading from 'Adolescent Eating Habits' to 'Adolescent Dietary Behaviors,' as the questions in the final checklist domains encompass both physical activity and daily routines. I believe this change enhances the clarity and accuracy of the terminology used (P.1-16, The terminology throughout the document has been updated.). I truly appreciate your insightful feedback, which has helped improve the precision of the terminology used in the survey instruments.

-Third Comment: I also agree with your third point that the frequency of consumption should vary depending on the specific food item being assessed. Using a single broad scale for all food items could potentially lead to inaccuracies in the collected dietary data and cause confusion when referencing the data in the future. To address this concern, I have added the full questionnaire as an appendix, which includes all questions and response options. This way, the frequency scales for different food items can be reviewed in detail (P.15, L.471-473, Appendix A2). Thank you for your valuable suggestion. Your input has greatly contributed to strengthening the validity and usability of the dietary assessment tool.

Reviewer 2 Report

Comments and Suggestions for Authors

The study presents an effort to develop a dietary adherence tool for adolescents in Korea, aligning with the Dietary Guidelines for Koreans. 

While the survey included 1,010 adolescents from 17 regions, the representativeness of the sample could be questioned. More information on the demographics (e.g., socioeconomic status, urban vs. rural distribution) would clarify whether the findings can be generalized across all adolescent populations in Korea.

The reliance on factor analysis and structural equation modeling is a good approach for validating the tool, but the study should provide more detail on the specific statistical methods used. This includes the criteria for item selection, how the 3 items were deemed unnecessary, and the rationale for the 6 new items added.

The work relies on self-reported dietary habits, which can be subject to bias. Adolescents may overestimate healthy behaviors or underestimate unhealthy ones. Incorporating objective measures, like food diaries or biomarker assessments, could enhance the validity of the findings.

While the tool aims to incorporate dietary culture, it may not fully account for the diverse cultural influences on eating behaviors among adolescents in different regions of Korea. Further exploring the socio-cultural dynamics and their impact on dietary choices would enrich the study.

The conclusion mentions promoting sustainable eating patterns, but the study does not elaborate on how the tool addresses sustainability beyond acknowledging family support. Practical strategies or interventions could be detailed to enhance the tool's applicability in promoting both health and sustainability.

The paper establishes a baseline for dietary adherence but does not discuss the potential long-term impact of using the tool. Future research should explore whether this tool can effectively track and influence dietary changes over time.

Author Response

Comments 2: The study presents an effort to develop a dietary adherence tool for adolescents in Korea, aligning with the Dietary Guidelines for Koreans. 

While the survey included 1,010 adolescents from 17 regions, the representativeness of the sample could be questioned. More information on the demographics (e.g., socioeconomic status, urban vs. rural distribution) would clarify whether the findings can be generalized across all adolescent populations in Korea.

The reliance on factor analysis and structural equation modeling is a good approach for validating the tool, but the study should provide more detail on the specific statistical methods used. This includes the criteria for item selection, how the 3 items were deemed unnecessary, and the rationale for the 6 new items added.

The work relies on self-reported dietary habits, which can be subject to bias. Adolescents may overestimate healthy behaviors or underestimate unhealthy ones. Incorporating objective measures, like food diaries or biomarker assessments, could enhance the validity of the findings.

While the tool aims to incorporate dietary culture, it may not fully account for the diverse cultural influences on eating behaviors among adolescents in different regions of Korea. Further exploring the socio-cultural dynamics and their impact on dietary choices would enrich the study.

The conclusion mentions promoting sustainable eating patterns, but the study does not elaborate on how the tool addresses sustainability beyond acknowledging family support. Practical strategies or interventions could be detailed to enhance the tool's applicability in promoting both health and sustainability.

The paper establishes a baseline for dietary adherence but does not discuss the potential long-term impact of using the tool. Future research should explore whether this tool can effectively track and influence dietary changes over time.

Response 2: Thank you very much for your thoughtful and constructive feedback. I sincerely appreciate the time and effort you invested in reviewing the manuscript. Below are the revisions made in response to your insightful comments:

Regarding your comment about the representativeness of the sample: I completely understand your concern. To clarify, the survey included 1,010 adolescents from 17 regions, and I have taken the population ratio between urban and rural areas into account. Specifically, 25% of the respondents were recruited from 'counties and below,' and the margin of error was set to < 0.04 at the 95% confidence interval to ensure the sampling met national statistical quality standards (P.3, L.100-103). Additionally, as per your request, I have newly included items of economic status in the survey instrument section (P.3, L.112) and responses in Table 1 (P.6, L.213). Thank you for raising this important point—your feedback has been incredibly helpful in reinforcing the clarity and comprehensiveness of the study.

Regarding your comment about the validation of the tool: I fully understand your concern. I followed the prescribed criteria for exploratory and confirmatory factor analysis (P.4, L.160-171). As a result of the analysis, three items were removed from the exploratory factor analysis because they did not align with theoretical expectations in the food intake domain (P.7, L.223-234), and one item (environmentally friendly products) was removed due to a strong correlation with the local food item in the dietary culture domain (P.9, L.252-254). In total, 4 out of 22 candidate items were removed, resulting in a revised tool with 18 items. I apologize for the mistake. I previously mentioned that 3 items were removed from the original 21 items in abstract, but I misspoke. In fact, 4 out of 22 candidate items were removed, resulting in a final set of 18 items. I have corrected this in the abstract to reflect the accurate numbers (P.1, L.24-25). Additionally, six new items were added based on expert feedback, as they felt that guidelines 1 ("Eat a balanced diet comprising grains, meat, fish, eggs, legumes, and dairy products every day"), 2 ("Eat less salty, less sugary, and less fatty food"), and 3 ("Drink plenty of water") were not adequately represented. The validity of the final scale was carefully reviewed based on the analysis results both before (18 items) and after incorporating expert feedback (24 items total) (P.9, L.273-282) and validated by comparing the candidate 18-item scale using the Independent Samples T-Test and Cohen’s Kappa Coefficient. I am happy to report that the validity of the final 24-item tool, which includes six additional items in the food intake domain to evaluate dietary guideline adherence, was maintained. Your comment has been instrumental in refining this section, and I sincerely appreciate your valuable insights. 

Regarding your comment about the possibility of self-reported bias among adolescents: Since all responses were based on adolescents' self-reports online, I have acknowledged the potential for bias. To minimize this, we provided them with standardized portion sizes in the food intake questionnaire to help them understand the amounts of food (P.15, L.471 Appendix A2). Additionally, I have incorporated the limitations of the self-reported nature of the tool, which may introduce bias, and discussed plans for future research to address these concerns (P.13, L.413-419). Your feedback has helped me ensure a more balanced discussion of the study’s limitations, and I truly appreciate your careful review.

Regarding your comment about the tool's incorporation of dietary culture: I analyzed whether there were significant differences in scale development items between respondents from urban and rural areas. However, no significant differences were found, so this analysis was not included in the manuscript. This may have led to the perception that regional dietary differences were not reflected. Nonetheless, I made efforts to incorporate key aspects of Korean dietary culture into the tool, such as questions about vegetables, including kimchi (Korean fermented vegetables), and spicy and salty foods (P.10, L.291, Table 5; P.15, L.471, Appendix A2), which are characteristic of Korean food flavors. For future research, it would be valuable to conduct a more detailed analysis that considers socio-cultural factors, such as regional differences in food accessibility. Your suggestion has provided valuable direction for future studies, and I truly appreciate your insightful perspective.

Regarding your comment about the need for more elaboration on how the tool addresses sustainability beyond family support: I recognize that further practical strategies are essential. In response, I suggest that future research focus on developing and evaluating concrete intervention programs, such as eco-friendly food purchasing guides or local food consumption promotion campaigns. These interventions would not only assist adolescents in adopting sustainable eating habits but also contribute to environmental sustainability, thus enhancing the tool’s applicability in promoting both health and sustainability over the long term (P.13, L.423-428). Your insightful feedback has been invaluable in expanding this discussion, and I sincerely appreciate your contribution to strengthening this aspect of the study.

Regarding your comment about the long-term impact of the tool: I have incorporated your suggestions into the manuscript, specifically acknowledging the need for future research to explore the long-term impact of the tool on dietary behavior. I have clarified that while the tool establishes a baseline for dietary adherence, its long-term effects remain unexplored, and I plan to focus on tracking and influencing dietary changes over time in future studies (P.13, L.416-423). Thank you again for your constructive input, which has greatly enhanced the clarity and depth of the study.

Round 2

Reviewer 1 Report

Comments and Suggestions for Authors

Thank you for addressing all the concerns raised. I just have some final points, based on your responses:

Line 112: please add here that height and weight were based on self reported data.

Through the new heading "Dietary behaviours" is not wrong, since you also include physical activity it might be more accurate to name this "Dietary and physical activity behaviours"

What do the authors mean by "exhilirating" exercise? Do they mean vigorous exercise?

Since processed foods in general were not questions, this needs to be addressed as a limitation in the discussion section. Also, since the frequency of responses in the questionnaire are all the same, this could be considered a limitation by experts in the nutrition field. Ideally, the responses should be tailored to the food item being questions as some foods are not eaten daily whereas others might be. The authors should address this in their work to be transparent because it means when taking these limitations into consideration, the accuracy of the conclusions might be impacted. So ideally, a follow on study would be needed to address these.

To circumvent these limitations somewhat, the authors can consider adding "pilot study" to their title. 

Author Response

comments 1: Thank you for addressing all the concerns raised. I just have some final points, based on your responses:

Line 112: please add here that height and weight were based on self reported data 

Through the new heading "Dietary behaviours" is not wrong, since you also include physical activity it might be more accurate to name this "Dietary and physical activity behaviours" 

What do the authors mean by "exhilirating" exercise? Do they mean vigorous exercise?

Since processed foods in general were not questions, this needs to be addressed as a limitation in the discussion section. Also, since the frequency of responses in the questionnaire are all the same, this could be considered a limitation by experts in the nutrition field. Ideally, the responses should be tailored to the food item being questions as some foods are not eaten daily whereas others might be. The authors should address this in their work to be transparent because it means when taking these limitations into consideration, the accuracy of the conclusions might be impacted. So ideally, a follow on study would be needed to address these.

To circumvent these limitations somewhat, the authors can consider adding "pilot study" to their title. 

Response 1: I am grateful for your comprehensive and insightful feedback. I deeply appreciate the time and attention you devoted to reviewing this manuscript. Your constructive suggestions have played a key role in enhancing the clarity and rigor of this study. Below are the specific revisions made in response to your comments. 

- First Comment: As per your suggestion, I have included a note that height and weight were derived from self-reported data following the BMI calculation (P.3, L.113-114). I appreciate your careful review and valuable input. 

- Second Comment: Thank you for your helpful suggestion. As you mentioned, I agree that "Dietary and physical activity behaviours" would more accurately reflect the content, considering the inclusion of physical activity (The terminology throughout the document has been updated). Thank you again for your valuable input.

- Third Comment: We originally asked about "exercise that makes you out of breath," which was translated as "exhilarating exercise." We agree that this may cause confusion, and we will revise it to "vigorous exercise" to more accurately reflect the intended meaning (The terminology throughout the document has been updated). Thank you for highlighting this point and helping us improve the clarity of our manuscript.

- Fourth Comment: Thank you for your question regarding processed foods in general. I would like to clarify that while general processed foods—such as processed beverages, snacks (e.g., chips), instant noodles, and frozen foods—were included during the pilot testing phase, their predictive value was found to be insufficient, leading to their exclusion from the final version of the tool. Only processed meats were initially retained, but even these items could not be further distinguished through factor analysis conducted during the nationwide survey, resulting in their eventual omission as well. While processed food items were not included in the final assessment tool, our research team fully acknowledges the importance of evaluating processed food consumption, and this issue has been comprehensively discussed as a limitation in the discussion section (P.13, L.413-421). I hope this explanation addresses your concern.

Additionally, I appreciate your feedback regarding the standardized frequency of responses in the questionnaire. While I designed the response frequencies based on an existing survey to maintain consistency and reduce respondent burden, I acknowledge that this approach may not fully capture the specific consumption patterns across different food groups. As such, I have also included this as a limitation in the discussion and noted that future studies should consider refining the frequency categories to better reflect the variability in dietary intake. Your insights have been valuable in improving the clarity and transparency of the manuscript (P.13, L.429-436).

- Fifth Comment: Thank you very much for your thoughtful suggestion. As previously mentioned, we have already conducted a pilot test in metropolitan areas in south korea during the first phase of the study. Given that the current paper focuses on a nationwide survey and the data collection process has already been completed, referring to this as a pilot study may not accurately reflect the scope of the research. I hope this explanation clarifies our decision. I appreciate your understanding and kindly ask for your consideration on this matter.